# Uncertain Resection in Lung Cancer: A Comprehensive Review of the International Association for the Study of Lung Cancer Classification

**DOI:** 10.3390/cancers17091386

**Published:** 2025-04-22

**Authors:** Xavier Cansouline, Abdelhakim Elmraki, Béatrice Lipan, Damien Sizaret, Mathieu Sordet, Anne Tallet, Christophe Vandier, Delphine Carmier, Myriam Ammi, Antoine Legras

**Affiliations:** 1Thoracic Surgery Department, Tours University Hospital, 37000 Tours, France; elmraki@gmail.com (A.E.); b.lipan@chu-tours.fr (B.L.); antoine.legras@chu-tours.fr (A.L.); 2N2C UMR 1069, University of Tours, INSERM, 37000 Tours, France; christophe.vandier@univ-tours.fr; 3Department of Pathology, Tours University Hospital, 37000 Tours, Franceanne.tallet@chu-tours.fr (A.T.); 4Department of Radiation Oncology, Tours University Hospital, 37000 Tours, France; m.sordet@chu-tours.fr; 5Department of Pneumology, Tours University Hospital, 37000 Tours, France; d.carmier@chu-tours.fr; 6Thoracic and Vascular Surgery Department, Angers University Hospital, 49000 Angers, France; myriam.ammi@chu-angers.fr

**Keywords:** NSCLC, oncology, resected lung cancer, uncertain resection, mediastinal lymphadenectomy, IASLC

## Abstract

In non-small cell lung cancer (NSCLC), the quality of surgical resection is assessed by the eighth TNM classification using the R criteria, sorting patients into complete (R0) or incomplete (R1-R2) tumor removal. In 2005, the International Association for the Study of Lung Cancer (IASLC) published a new classification, including a fourth category called “uncertain resection [R(un)]” to spot high-risk patients for recurrence despite resection usually considered as complete. This classification is little used, and its relevance in clinical practice remains unclear.

## 1. Introduction

The treatment of early and locally advanced stage lung cancer primarily relies on surgical resection with the intention to remove all disease [1,2,3]. The quality of resection is assessed by the R criterion of the UICC TNM classification [4], which is divided into three categories: complete resection (R0), microscopic residue (R1), and macroscopic residue (R2). This criterion is a major prognostic factor and can alone determine access to adjuvant treatments [5,6]. However, a significant portion of resected patients classified as R0 experience recurrence and succumb to their disease, despite well-conducted surgery and apparently complete resection [7,8]. To better identify these at-risk patients, the IASLC proposed a fourth category for the R criterion in 2005: uncertain resection [R(un)] [9]. It includes patients who do not present proven residual tumor but for whom there is doubt about disease persistence. An uncertain resection is defined as a resection with margins proved to be free of disease microscopically, but the intraoperative lymph node evaluation is less rigorous than systematic nodal dissection or lobe-specific systematic nodal dissection and/or the highest mediastinal node removed is positive (HMLN+) and/or the bronchial margin shows carcinoma in situ (CIS BRM) and/or the pleural lavage cytology is positive (R1 Cy+).

According to the IASLC classification, lobe-specific systematic nodal dissection implies retrieving hilar nodes and, at least, three mediastinal nodal stations depending on the lobar location of the primary tumor, always including subcarinal nodes [7]. For the right upper and middle lobes, this includes two of the following three stations: superior paratracheal [2R], inferior paratracheal [4R], and pretracheal [2-4R]. For the right lower lobe, this includes right inferior paratracheal nodes [4R] and either the paraoesophageal [8] or pulmonary ligament nodes [9]. For the left upper lobe, this includes subaortic [5] and anterior mediastinal nodes [6]. For the left lower lobe, this includes paraesophageal [8] and pulmonary ligament nodes [9]. The lymph node specimen should include at least six nodes: three removed from N1 stations and three removed from N2 stations.

This classification, which was released almost twenty years ago, was intended to harmonize the criteria for resection quality on an international scale and to identify a new prognostic group of patients for whom local disease control appears uncertain. However, this classification still seems to be underutilized in practice compared to the classic R0-1-2 triad. As the ninth edition of the TNM is being prepared, an assessment seems necessary. Here, we propose a literature review to evaluate its relevance and its impact on overall survival and disease-free survival, focusing on R(un), which provides arguments in favor of a higher risk of recurrence among patients previously classified as R0.

## 2. Materials and Methods

We conducted a comprehensive literature review and synthesized the results of all published series evaluating the prognostic value of R(un) according to the IASLC or one of its components. Series were searched on PubMed, Cochrane, MEDLINE, and Google Scholar. We manually searched the keywords “lung cancer” and IASLC”, combined with the connector “AND”, and additional keywords “uncertain resection”, “highest node”, “pleural cytology”, and “carcinoma in situ”. The PRISMA flowchart in Figure 1 shows the selection process for the 68 studies that were selected.

For more clarity, we deliberately omitted data comparing incomplete resection groups, as the prognostic impact of the latter is no longer in doubt.

## 3. Results

We first described the literature background of the IASLC classification; then, we reviewed validation studies of R(un) and examined the impact of HMLN+, CIS BRM, and Cy+ separately.

### 3.1. Roots of the IASLC Classification

To better understand the relevance of the IASLC classification, it is first necessary to examine the literature upon which it relied to define R(un).

Naruke et al. [10] were the first, in 1978, to mention a “relatively curative” resection in which “all grossly visible tumor had been removed”. They indeed considered that involvement beyond the visceral pleura or the presence of mediastinal lymph node metastases did not allow for a conclusion of complete removal of the tumor. Furthermore, radical lymph node dissection and clear resection margins were required. They demonstrated that N1 involvement (hilar, stations 10 to 12) or N2 involvement (mediastinal, stations 1 to 9) was associated with a 5-year survival rate lower than N0 patients (limited to intraparenchymal nodes, stations 13 and 14). Their landmark publication fulfilled its objective of establishing the prognostic value of mediastinal lymph node involvement while proposing a lymph node mapping still used to this day; however, it did not dwell on this definition of uncertain resection, which was more semantic at the time.

In 1998, the Spanish Society of Pulmonology and Thoracic Surgery (SEPAR) published recommendations for standardizing the diagnosis and classification of lung cancer [11] and incorporated the work of Mountain [12] to define “presumed incomplete” surgery. It applies when margins are clear but with one of the following elements: the absence of radical lymph node dissection, the farthest positive lymph node (which can be the highest but also the lowest, in zone 9), if dissection was radical, and the presence of pleural effusion with positive cytology without tumor involvement of the pleura. The authors also proposed classifying atypical resections (wedge resections) as presumed incomplete surgeries. This definition is very similar to that proposed by the IASLC, differing only in the absence of margin abnormalities and the notion of positive pleural cytology without tumor involvement of the pleura. If carcinoma in situ at the margins is simply not mentioned, positive pleural cytology is accepted only in the absence of pleural involvement by the tumor. This is because the authors believed it could be desquamations from the tumor or from lymphatics and/or peripheral lymph nodes, consisting of tumor cells incapable of migrating or causing pleural metastases. In the case of pleural invasion by the tumor, the same cytology was then considered malignant, and the surgery was deemed incomplete.

Regarding the CIS BRM component, it is based on the work of Snijder et al. [13] and Massard et al. [14], who showed that carcinoma in situ at the bronchial margin primarily implied significant recurrence without demonstrating an effect on survival. However, these are retrospective series with very small sample sizes, thus having a limited power level. Given the natural progression of these lesions [8,15], it is legitimate to consider the presence of carcinoma in situ at the bronchial resection margin as a potential poor prognostic factor.

Regarding the Cy+ component, numerous studies had already demonstrated that positive cytology in pleural lavage was an independent factor for poor prognosis [16,17,18], often associated with adenocarcinomas [19] of more advanced stages [20,21,22,23], with pleural exposure [24,25,26], N2 involvement [27], or vascular invasion [20]. All these publications, even with small sample sizes, showed lower survival and higher recurrence rates in Cy+ patients.

### 3.2. Validation Studies

After the publication of the classification in 2005, several authors sought to verify the clinical relevance of the classification. These series are synthesized in Table 1 and Figure 2. To date, only retrospective series can be found, sometimes with significant sample sizes:

Gagliasso et al. [28] were the first to investigate uncertain resection by publishing a retrospective series of all patients who underwent surgery at their center in Italy over a period of 10 years and by reclassifying the R criterion according to the IASLC. Overall, 185 patients (15%) out of 1277 patients were reclassified as R(un); in 97% of the cases, this reclassification was due to inadequate lymph node dissection (107/185, 56.6%) or involvement of the highest mediastinal lymph node (76/185, 41%). Survival analyses found a higher mortality rate in the R(un) group compared to the R0 group [hazard ratio (HR) 1.69, 95% CI: 1.412–2.024, *p* = 0.0001] with a significantly decreased 5-year survival rate (37.3% vs. 58.8%, median survival 39.9 months vs. 80.1 months). It is noteworthy that the patients classified as R(un) had more often undergone atypical resection than those in the R0 group (15% vs. 1.3%), which could be a confounding factor, as wedges are associated with a poorer prognosis than anatomical resections. Nevertheless, after their multivariate analysis, R(un) remained a significant predictor of mortality (adjusted HR 1.352, 95% CI: 1.101–1.661, *p* = 0.004). In more detail, it can be noted that the HMLN+ patients (n = 76) had a 5-year survival rate of 28.8%; for the CIS BRM patients (n = 5), the 5-year survival rate was 40%, and for those with inadequate dissection (n = 107), it was 44.2%.

In 2019, Osarogiagbon et al. [30] published a retrospective series based on a population of 12 American hospitals over 10 years, totaling 3361 patients. Their study revealed that 63.8% of the R0 patients, according to UICC definitions, had to be reclassified as R(un), that is, 2044 cases, almost always due to inadequate lymph node dissection (98%). They once again reported higher mortality among R(un)-classified patients (adjusted HR 1.36, 95% CI 1.19–1.56, 5-year OS 54% vs. 64%, median OS 69 months vs. not reached). This relationship persisted in the N0 (HR 1.31, 95% CI: 1.12–1.53, median OS 83 months vs. not reached) and N+ (HR 1.24, 95% CI: 0.94–1.62, median OS 44 vs. 62 months) subgroups. It is also notable that the HMLN+ factor seemed to emerge as a more significant mortality factor than inadequate dissection (HR 1.5, 95% CI: 1–2.31), without statistically differing from the R0-N2 group (HR 1.08, 95% CI: 0.71–1.67). Finally, the authors suggested that the absence of mediastinal dissection and complete absence of lymph node sampling would be leading mortality factors (HR 1.46, 95% CI: 1.24–1.73 and HR 1.79, 95% CI: 1.45–2.22, respectively).

Yun et al. [31] evaluated the IASLC classification, focusing on pN2 patients, with a retrospective South Korean cohort of 1039 subjects. Among the R0 patients, 206 (21.8%) were reclassified as R(un), exclusively for HLMN+ (89.8%) or inadequate lymph node dissection (10.2%), and 6 patients transitioned from R1 to R(un) due to CIS BRM. They reported inferior survival of the R(un) group compared to the R0 group in a univariate analysis (HR: 1.25, 95% CI: 1.01–1.56, *p* = 0.04, median OS 55 vs. 71 months, 5-year OS 45.8% vs. 54.7%), but this difference faded in a multivariate analysis, notably considering N2 subdivisions (HR: 1.06, 95% CI: 0.85–1.34, *p* = 0.595). Indeed, the R(un) patients had significantly more N2a2 and N2b involvement, which had already been identified as mortality factors. It is also noteworthy that the HMLN+ factor appeared to have a significantly worse prognosis than inadequate lymph node dissection (median OS 54 vs. 97, 5-year OS 45.4 vs. 64.6, *p* = 0.07), but these data should be interpreted cautiously given the small sample size of the inadequate lymph node dissection group (n = 21). There were no data about CIS BRM patients.

Wang et al. [32] conducted a similar study on a local Chinese series of 2782 R0 patients according to the UICC. After their review, 885 (32%) were reclassified as R(un), among which only inadequate lymph node dissections (81%) and HMLN+ (19%) were found. Based on their multivariate analysis, the authors reported a better survival of R0 (adjusted HR: 1.302, 95% CI: 1.091–1.555, *p* = 0.003) and also a better recurrence-free survival (adjusted HR: 1.177, 95% CI 1.045–1.380, *p* = 0.031). Within the R(un) group, HMLN+ seemed to present lower OS and DFS compared to inadequate lymph node dissection (5-year OS~38% vs. 75%, *p* < 0.001, based on a Kaplan–Meier curve) (5-year DFS~25% vs. 72%, *p* < 0.001, based on a Kaplan–Meier curve). In their subgroup analysis, R(un) negatively impacted the survival and DFS of the N0 (OS *p* = 0.002, DFS *p* = 0.033), N2 (OS and DFS *p* = 0.008), and stage III patients (OS *p* = 0.002, DFS *p* = 0.003) but not the N1 patients (OS *p* = 0.79, DFS *p* = 0.76) nor the stage I (OS *p* = 0.037, DFS *p* = 0.276) and stage II (OS *p* = 0.187, DFS *p* = 0.122) patients. Finally, the authors reported that segmentectomy, thoracotomy surgery, and left-sided localization would be risk factors for R(un) occurrence.

Ren et al. [33] published a retrospective Chinese series of 5293 patients treated over a period of 4 years. Once again, they found 1371 patients reclassified as R(un) from the initial R0 group, accounting for 25.9% of its population, almost exclusively due to inadequate lymph node dissection (67.7%) or HMLN+ (37.2%). They reported lower OS (adjusted HR 1.41, 95% CI: 1.28–1.54, median DFS 37 vs. 57 months, *p* < 0.001), and DFS (adjusted HR 1.52, 95% CI: 1.38–1.67, 5-year OS 46% vs. 71%, median OS 35 months vs. not reached, *p* < 0.001) among R(un) patients based on a multivariate model. While HMLN+ involvement was presented as an independent prognostic factor for DFS (adjusted HR 1.21, 95% CI: 1.06–1.39, *p* = 0.003) and OS (adjusted HR 1.23, 95% CI: 1.07–1.42, *p* = 0.005), its effect seemed weaker than R(un) N+ HMLN- involvement (adjusted HR DFS 1.37, 95% CI 1.1–1.71, *p* unavailable; adjusted HR OS 1.31, 95% CI 1.06–1.61, *p* unavailable). It is worth noting that in their study, all patients had undergone pleural lavage cytology, which was positive for only 11 patients.

Kadomatsu et al. [34] published a Japanese series of 355 R0 patients according to the UICC, of which 158 (44.5%) were reclassified as R(un), with 137 (87%) due to inadequate lymph node dissection. Survival figures for the overall R(un) group were not provided. However, there was no reported difference in survival within the stage I, stage II–III, and N0 subgroups. Only the N+ group differed, with lower survival in the R(un) group (adjusted HR 2.657, 95% CI: 1.197–5.899, *p* = 0.007; 5-year OS 43.4% vs. 65%). However, the low sample sizes of 39 and 26 patients for these subgroups suggest that caution is required when interpreting these figures. No difference in DFS was reported by the authors.

Recently, Lee et al. [35] presented a South Korean series of 910 patients, including only clinical stage IIIB-N2 patients who received neoadjuvant radio-chemotherapy followed by surgery. Here, 329 patients were reclassified as R(un) due to inadequate lymph node dissection (84.5%) and/or HMLN+ (31.6%). It is also interesting to note that 245 patients (28% of the UICC R0 population) were also reclassified as incomplete resection, mainly due to extracapsular lymph node invasion (ECE, 77%) and/or positive known nodes not removed (28.2%). Thus, a cohort initially composed of 96.3% UICC R0 found itself in substantially comparable proportions between R0 (33.2%), R(un) (36.2%), and incomplete R (30.7%) according to the IASLC. No significant difference was found in the R(un) group compared to the R0 group regarding OS (adjusted HR 1.22, 95% CI: 0.95–1.56, *p* = 0.114) or DFS (adjusted HR 1.25, 95% CI: 0.98–1.59, *p* = 0.113), although there was a trend toward poorer outcomes. Not surprisingly, the R1/2 patients showed markedly lower OS (adjusted HR 1.54, *p* = 0.001) and DFS (adjusted HR 1.51, *p* = 0.001) compared to the R0 group. Proven mediastinal involvement (ypN+) seemed to be predictive of recurrence (adjusted HR 1.31, 95% CI: 0.96–1.8, *p* = 0.071) but not of death (adjusted HR 1.13, 95% CI: 0.81–1.57, *p* = 0.476), as did HMLN+ (adjusted DFS HR 1.18, 95% CI: 0.92–1.5, *p* = 0.183; adjusted OS HR 0.77, 95% CI: 0.59–1.02, *p* = 0.196). However, these results were difficult to interpret since the absence of significant differences could reflect the effect of neoadjuvant radio-chemotherapy, which may result in mediastinal downstaging, as observed in each group of the study (R0: 54% N0; R(un): 41.6% N0; R1/2: 14.3% N0), which included only cN2 patients. This downstaging could decrease the proportion of occult N+ in the R(un) group, thus bringing it closer to the R0 group and blurring the survival differences usually observed. Finally, it is noteworthy that the authors reported significantly better survival in the IASLC R0 group compared to the UICC R0 group (*p* = 0.011), which supported the idea of significant heterogeneity within the latter group.

Wen et al. [36] also described a series of 5200 patients, of whom 1727 were reclassified as R(un) (33%) due to insufficient LND (68.3%) or HMLN+ (38.3%) (CIS BRM = 3). They reported a lower OS (adjusted HR 1.4, 95% CI: 1.223–1.603, *p* < 0.001) and DFS (adjusted HR 1.28, 95% CI: 1.151–1.434, *p* < 0.001) in the R(un) group. In their subgroup analysis, R(un) remained relevant for survival and DFS for all parameters studied (age, sex, smoking, histology) except for radiographic appearance and stage. It seemed that R(un) with ground-glass opacities (HR 1.19, 95% CI: 0.78–1.81, *p* = 0.4) or stage I disease (HR 0.88, 95% CI: 0.68–1.14, *p* = 0.34) survived as well as their R0 counterparts. The DFS of R(un) with ground-glass opacities was comparable (HR 1.19, 95% CI: 0.88–1.62, *p* = 0.29), while a protective effect of R(un) was found in stage I (HR 0.81, 95% CI: 0.61–0.98, *p* = 0.024). However, it should be noted that stage I and ground-glass opacities were predominantly reclassified as R(un) due to insufficient LND, whereas stages II-III and solid nodules were more likely to be reclassified due to HMLN+. Finally, the authors showed that the HMLN+ patients had lower OS than the patients with insufficient LND (*p* < 0.0001).

Chen et al. [37] focused on patients who underwent lobectomy with bronchial resection–anastomosis. They reported a series of 682 patients, of whom 631 were classified as R0 UICC, reclassified as 489 R0 (71.7%), 110 R(un) (16.1%), and 83 R1-2 (12.2%). The authors observed a lower DFS (adjusted HR 1.59, 95% CI: 1.09–2.31, *p* = 0.023) and overall OS (adjusted HR 1.54, 95% CI: 1.02–2.33, *p* = 0.040) in the R(un) group compared to R0. In their subgroup analysis, this difference was observed in the N2 patients (5-year DFS 29.1% vs. 59.1%, adjusted *p* = 0.010; 5-year OS 43.5% vs. 56.6%, adjusted *p* = 0.010). These findings need to be considered alongside the fact that 82 out of the 110 patients in the R(un) group were reclassified as HMLN+, indicating N2 status. Consequently, the N0 and N1 groups comprised only 16 and 5 patients, respectively, limiting the reliability of the results.

In 2025, a French study [38] reported R(un) as a worsening factor for OS (adjusted HR 1.26; 95% CI: 1.03–1.52, *p* < 0.001) and DFS (adjusted HR 1.23; 95% CI: 1.03–1.46). The subgroup analyses among R(un) patients suggested HMLN+ and very poor LND (less than three lymph nodes or station 7 not examined) as a significant factor for low OS [HRs 3.35 (95% CI: 2.05–5.49) and 2.17 (95% CI: 1.39–3.38), respectively] and DFS [HRs 2.66 (95% CI: 1.71–4.15) and 2.05 (95% CI: 1.35–3.10), respectively].

Liu et al. [39] studied the impact of R(un) when associated with the new IASLC tumor gradation of adenocarcinomas [40], which mainly relies on anatomo-pathologic characteristics. They found that R(un) was associated with poor OS (adjusted HR 1.57, 95% CI: 1.20–2.05, *p* = 0.001) and RFS (adjusted HR 1.73, 95% CI: 1.37–2.19, *p* < 0.001) in the overall population but also showed that R(un) reduced OS and RFS in only in aggressive grade 3 tumors (OS 75.3% vs. 56.5%, *p* < 0.001; RFS 68.3% vs. 41.9%, *p* < 0.001).

### 3.3. The “Highest Mediastinal Lymph Node” Component

Some authors focused on the prognostic impact of the involvement of the highest lymph node, a criterion casting doubt on the persistence of lymphatic metastases despite well-performed mediastinal dissection. These series are summarized in Table 2 and Figure 3.

Sakao et al. [41] published a series of 53 N2 patients, including 14 HMLN+ who were naive to any neoadjuvant and adjuvant therapy. They reported a markedly lower survival among the HMLN+ patients (adjusted HR 3.225, 95% CI: 1.153–9.009, *p* = 0.026). However, the dissection was carried out very aggressively, with HMLN defined as being located above the innominate vein trunk (stations R2 and L2), and the authors even performed a median sternotomy for left lung tumors and harvested nodes below the thyroid. These locations are rarely sampled in common practice and ultimately resemble N3 involvement, which could explain, in this small-sized series, the marked difference in survival among the HMLN+ patients, some of whom could have occult N3.

Zheng et al. [42] reported, in their retrospective series of 549 patients, including 246 HMLN+, a similarly reduced survival among the latter compared to HMLN- patients (adjusted HR 1.584, 95% CI: 1.217–2.062, *p* < 0.0001; 5-year OS 13% vs. 29%, median OS 24.43 vs. 36.48 months, *p* < 0.0001). Here, the dissection was carried out up to stations 2R and 4L, mostly outside the TABC, according to the article’s diagrams, representing a more conventional approach to dissection than in Sakao et al.’s series. However, it is notable that skip N2 HMLN+ had significantly more severe mediastinal involvement than skip N2 HMLN- since the number of mediastinal lymph node stations involved was higher in the former group (*p* < 0.0001). While the number of skip N2 stations involved seemed to be a mortality factor in this study (*p* < 0.0001), its effect was not found in a multivariate analysis (adjusted HR 0.945, 95% CI 0.887–1.213, *p* = 0.224), reinforcing the impression of an effect primarily linked to HMLN+.

Park et al. [43] reported a retrospective series of 339 N2 patients, among whom 142 (42%) were HMLN+. Here, the authors differentiated between patients with the highest involved lymph node (2R or 4L, 5 or 6) and those with the farthest involved lymph node (either the highest or the lowest in station 9). The results were similar in both groups, with a similar survival to the control group: adjusted HR 1.015 (95% CI: 0.751–1.371, *p* = 0.924), adjusted HR 1.050 (95% CI: 0.775–1.423, *p* = 0.755), respectively.

In a retrospective series of 266 pT1-4N2M0 patients, of whom 128 were HMLN+ (stations 2R, 4L, 5, or 6), Wang et al. [44] found no difference in survival (adjusted HR 1.00, 95% CI: 0.702–1.428, *p* = 0.99). It is notable here that there was a higher mortality among the patients who received neoadjuvant therapy (adjusted HR 1.88, 95% CI 0.928–3.834, *p*= 0.079), which is consistent with more severe preoperative involvement and thus poorer prognosis.

Recently, Marziali et al. [45] published a retrospective series of 68 cN0, pN2 patients, with 31 HMLN+, reporting a lower OS (adjusted HR 2, 95% CI: 0.7–6.1, *p* = 0.002) and inferior DFS (adjusted HR 3.2, 95% CI: 1.4–7.4, *p* = 0.008) in the R(un) patients. Here, the definition of HMLN was different from other studies, as lymph node dissection was sometimes performed in a lobe-specific manner. Thus, out of the 31 patients, 6 were HMLN+ in station 7: 2 for tumors of the left lower lobe and 4 for tumors of the right lower lobe. The latter theoretically did not correspond to an HMLN+ but rather to inadequate dissection, as it did not include station 4R.

Liu et al. [46] also published a retrospective series of 468 pT1-4N2 patients, including 219 HMLN+. They reported decreased OS (adjusted HR 1.45, 95% CI 1.07–1.99, *p* = 0.017) and recurrence-free survival (adjusted HR 1.26, 95% CI 0.94–1.68, *p* = 0.0115) in the HMLN+ patients. This result was reinforced in the subgroup of stage IIIA patients (384 patients, 82.1% of the total cohort) for OS (adjusted HR 1.70, 95% CI 1.19–2.42, *p* = 0.003) and recurrence-free survival (adjusted HR 1.46, 95% CI 1.06–2.02, *p* = 0.002). These differences were not found in the IIIB patient subgroup, probably because the severity of the disease at this stage overshadowed the prognostic impact of HMLN.

**Table 2 cancers-17-01386-t002:** Published series regarding the highest mediastinal lymph node.

			Distribution (%)			
Author, Year, Country	Patients (n)	Design	HMLN-	HMLN+	HR DFS (*p*)	HR OS (*p*)
Wang et al. [44], 2021, China	pT1-4N2M0 (266)	Retrospective, Single-Institution	138 (52%)	128 (48%)	NA	1(0.993)
Sakao et al. [41], 2006, Japan	pT1-3N2M0 (53)	Retrospective, Single-Institution	39 (74%)	14 (26%)	NA	3.225 (0.026)
Park et al. [43], 2019, South Korea	pT1-4N2M0 (339)	Retrospective, Single-Institution	197 (58%)	142 (42%)	1 (0.99)	1.015 (0.924)
Marziali et al. [45], 2023, Italy	pT1-4N2M0 (68)	Retrospective, Single-Institution	37 (54%)	31 (46%)	3.2 (0.008)	2 (0.002)
Zheng et al. [42], 2010, China	pT1-4N2M0 (549)	Retrospective, Single-Institution	303 (55.2%)	246 (44.8%)	NA	1.584 (<0.0001)
Liu et al. [46], 2024,China	pT1-4N2M0 (486)	Retrospective, Single-Institution	249 (53.2%)	219 (46.8%)	1.26 (0.115)	1.45 (0.017)

### 3.4. The “Carcinoma In Situ at Bronchial Margin” Component

To date, there are very few series on CIS BRM. Indeed, the occurrence of this criterion remains extremely rare, even in large retrospective series [29,33]. There are about ten series [47,48,49,50,51,52,53,54,55,56,57] in which CIS BRM is reported in very small numbers, as shown in Table 3. Statistical tests are not available, but it is noted that the reported cases are almost always associated with a primary tumor of squamous cell carcinoma type. Ultimately, no author has been able to conclude a negative influence of CIS BRM on survival.

In 2017, Lee et al. [57] published a cohort of 1800 patients, including 18 with CIS BRM and 42 with extra-mucosal involvement of the bronchial margin (R1-EMD). The primary tumors of the CIS BRM patients were predominantly squamous cell carcinomas (88.9%). The authors reported a higher local recurrence rate of CIS BRM compared to R0 (*p* = 0.008) without it affecting survival (adjusted HR 0.985, 95% CI: 0.373–2.602, *p* = 0.98).

### 3.5. The “Pleural Cytology” Component

Regarding Cy+, nearly twenty series, mainly retrospective, have been published after the release of the IASLC classification [58,59,60,61,62,63,64,65,66,67,68,69,70,71,72,73,74,75,76]. Although these series were not developed for the purpose of validating the IASLC classification, they constitute the dense literature in favor of this criterion for R(un). On average, Cy+ patients represented 5% of the total sample size, and most series proposed pre-resection lavage (prePLC), performed just after thoracic access, sometimes followed by post-resection lavage (postPLC) before thoracic closure.

A meta-analysis conducted by Wang et al. [77] in 2016, comprising 28 studies with a total of 28,714 patients, including 1434 cases of Cy+, revealed that patients with positive pleural lavage cytology (PLC+) pre- and post-resection had significantly higher mortality compared to patients with negative cytology (PLC−): HR 2.89 (95% CI: 2.48–3.37, *p* < 0.00001) and HR 2.70 (95% CI: 1.90–3.83), respectively. This effect was reinforced when focusing on early stage I, where prePLC+ had an HR of 3.29 (95% CI: 2.55–4.25, *p* < 0.00001) and postPLC+ had an HR of 4.85 (95% CI: 2.31–10.20, *p* < 0.0001). Similar figures were found concerning recurrence (all stages combined), where prePLC+ had an RR of 2.45 (95% CI: 1.91–3.15) and postPLC+ had an RR of 2.37 (95% CI: 1.11–5.09). Additionally, the authors also reported a significantly increased risk of pleural recurrence (RR 4.84, 95% CI: 3.03–7.73, *p* < 0.00001) and distant recurrence (RR 2.62; 95% CI: 1.72–4, *p* < 0.00001) in the prePLC+ group. A similar effect was found in the postPLC+ group on pleural recurrence (HR 3.39, 95% CI: 1.26–9.10, *p* = 0.02) but not on distant recurrence (HR 1.7, 95% CI: 0.8–3.62, *p* = 0.17). However, the numbers of events and patients in this group were much lower than in the prePLC+ group, which could mask results due to a lack of power.

Subsequent series have been published, yielding substantially similar results: Nakamura et al. [76] reported a prognostic contribution of postPLC+ (adjusted HR 2.12, 95% CI: 1.19–3.77, *p*= 0.01) but not of prePLC+ (adjusted HR 1.09, 95% CI: 0.56–2.12, *p* = 0.79) with a simplified lavage of 20mL saline solution. Onodera et al. [75] described a decreased DFS in both groups (5-year DFS prePLC+ 26.7% vs. 76.9%, *p* < 0.0001; 5-year DFS postPLC+ 14.3% vs. 76%, *p* < 0.0001), which was not significant in a multivariate analysis (HR prePLC+ 1.82, *p* = 0.19 and HR postPLC+ 2.03, *p* = 0.14). Mizuno et al. [74] also showed decreased OS and DFS in PLC+ patients (5-year OS 61.5% vs. 81.7%, *p* < 0.001; 5-year DFS 31.1% vs. 75.7%, *p* < 0.001), persisting in a multivariate analysis only for DFS (adjusted HR OS 1.34, 95% CI: 0.80–2.13, *p* = 0.25; adjusted HR DFS 1.70, 95% CI: 1.12–2.47, *p* = 0.013). These series are all subject to the same limitations, namely, their retrospective nature and very small sample sizes in the PLC+ groups.

Very recently, Recuero Diaz et al. [78] published a multicenter prospective series of 684 patients, of which 15 were PLC+. These patients showed a higher prevalence of advanced stages (pIIB-IIIB, *p*= 0.02), pleural (*p* = 0.005) and vascular invasion (*p* = 0.02), N+ involvement (*p* < 0.001), and larger tumor sizes (*p* = 0.005) with pT3-4 (*p* = 0.09) and poorly differentiated histology (*p* = 0.003). In their multivariate analysis, PLC+ emerged as the major prognostic factor for recurrence (OR 3.46, 95% CI: 2.25–5.36, *p* < 0.001), surpassing N+ involvement, pTNM stage, and poorly differentiated histology, among others.

Therefore, Cy+ is a very strong independent prognostic factor, indicating both a tumor that is often more advanced or aggressive and favoring early recurrence and poor survival, placing patients in risk groups close to advanced stage III [26], even at stage I [71]. To account for this factor, some authors propose upgrading the tumor criterion (T) and considering adjuvant treatment [69,74], especially in early stages that may be falsely reassuring.

## 4. Discussion

Most published series found a negative prognostic impact of R(un) status. Uncertain resection according to the IASLC 2005 allows for the individualization of a group of patients with intermediate prognosis, worse than R0 but better than R1-2 [28,29], and, therefore, emerges as a relevant criterion that should raise concerns about early recurrence and increased mortality. However, various studies highlight that R(un) encompasses a heterogeneous population, and the prognostic strength of R(un) is often inferior to standard criteria such as TNM stage or N+ involvement.

From an epidemiological perspective, Wang et al. [32] reported that segmentectomy, thoracotomy, and left-sided tumors are risk factors for the occurrence of R(un). These are likely indirect markers: lymph node dissection may be less extensive in segmentectomy cases, as it is considered less crucial in patients with small tumors. The authors considered that the left side was more challenging for dissection due to nearby structures (thoracic duct, aortic arch, recurrent nerve…). Finally, thoracotomy is applied to patients with more advanced tumors, likely at a higher risk of lymph node invasion and thus positive mediastinal lymph nodes and HMLN+. For HMLN+, risk factors also include cN+ involvement and severe mediastinal involvement (non-skip N2 and multiple N2) [41]. Cy+ appears to be favored by pleural and vascular involvement, occurring more frequently in larger tumors, with pN+ involvement, and low differentiation [78].

Furthermore, R(un) is a composite of several criteria, themselves unequal. While the prognostic role of insufficient dissection and HMLN+ has been well-documented, CIS BRM remains a rare event. A review conducted on the subject by Vallières et al. [79] did not identify an effect on survival, although local recurrence seemed to be slightly increased. The prognostic strength of this criterion remains to be determined, but it should be noted that CIS BRM is classified as R1 according to the UICC. Its migration to R(un) seems, therefore, justified, pending further investigation to determine if this criterion indeed excludes a complete resection. As CIS BRM is located in the inner face of bronchial airways, it would be consistent to consider reinforced endoscopic surveillance for those patients.

The R(un) group is predominantly composed of patients who have not undergone sufficient lymph node dissection, making it a melting pot of true N0, true and occult N1, and true and occult N2, with proportions impossible to specify. Thus, under-evaluated patients may not benefit from appropriate adjuvant therapies or surveillance, resulting in increased mortality in R(un). R(un) is also a good example of the Will Rogers phenomenon [80,81], whereby the prognosis of the R0 group improves due to the migration of cases with a poorer prognosis to the new R(un). This was also illustrated by Lee et al. [35], with R0 IASLC survival being better than that of the UICC R0 group (*p* = 0.011).

A portion of the R(un) group is also represented by patients with the highest mediastinal lymph node involvement, which represents advanced N2 involvement and seems to have even more prognostic value when lymph node dissection is extensive [41,42]. This criterion is difficult to analyze as it heavily depends on the quality of dissection and must also consider N2 subdivisions, particularly the possible presence of skip N2, which notably affects prognosis [82,83]. Additionally, it is challenging for both the surgeon and pathologist to precisely identify the highest lymph node in the chain, as dissection is usually performed en bloc with mediastinal fat to limit the risk of rupture. As a result, most authors report the highest station involved, raising doubts about the possibility of residual involvement by contiguity but still providing an assessment of mediastinal involvement, as HMLN+ is often associated with multiple N2 involvement, the poor prognosis of which is increasingly documented [84,85]. This highlights the value of orienting lymph node dissection during surgery to properly mark the highest node removed.

On the contrary, the Cy+ criterion, although very rare in cases where pleural lavage is performed, appears to be a leading factor in mortality and recurrence. Its occurrence is more likely when the tumor invades the visceral pleura [78], raising the question of a pathophysiological continuum between this lavage cytology and the development of carcinomatous pleural effusion, classified as M1a. Nevertheless, Mizuno et al. [73], by performing intraoperative lavage analysis, showed that Cy+ patients had better survival than M1a patients (*p* < 0.001), while also detecting pleural dissemination in 9% of the positive cases. This criterion is particularly relevant as it may affect tumors that solely invade the visceral pleura and are therefore classified as T2, for which adjuvant treatments are not routinely offered. Cy+ is thus a very strong independent prognostic factor, indicating both a more advanced or aggressive tumor, placing patients in risk groups akin to advanced stage III [26], even at stage I [74]. To account for this factor, some authors propose upstaging the T criterion of the tumor [67] and considering adjuvant treatment [69,74], especially in early stages that might appear falsely reassuring. Ogawa et al. [86] showed improved OS and DFS in patients who received adjuvant chemotherapy, which should encourage us to perform pleural lavage cytology. The question of whether those patients should receive adjuvant chemotherapy similarly to locally advanced stages or be treated as metastatic disease needs to be explored.

R(un) appears to be less discriminatory in stage I and ground-glass opacities [32,87,88]. This could be explained by a low probability of lymphatic invasion for these highly localized tumors [89]. Thus, Lee et al. [87] showed that less extensive dissection in stage I and ground-glass opacities was not associated with poorer prognosis. This could explain the lack of difference in these subgroups [32,34,36,37]. In 2011, Darling et al. published the ACOSOG Z0030 trial [90], which demonstrated no difference in survival between systematic sampling (limited to one lymph node per area), pN0 confirmed intraoperatively, and radical lymph node dissection in cT1-2, N0 or non-hilar N1 patients. This finding has led some surgeons to debate the utility of lymph node dissection in early stages [91] and propose sampling for small peripheral tumors in cN0 [92]. This conservative approach, although interesting, requires perfect coordination and availability between the surgeon and pathologist, which can be challenging in routine practice. An alternative could be the use of intraoperative indocyanine lymphatic mapping [93,94], for example, to reduce the number of intraoperative analyses, but this technique needs more investigations and proof of reliability.

The prognostic significance of R(un) becomes evident when advancing beyond stage IIb, with larger tumors and a higher probability of lymphatic dissemination. For patients classified as N0, R(un) primarily reflects the quality of lymph node dissection, which is likely inadequate given the advancement of the disease. The imprecision of lymph node dissection may represent a missed opportunity for these patients due to the persistence of potentially removable residual disease (especially skip N2) and understaging. For N1 and N2 patients, R(un), because of inadequate dissection or HMLN+, should also raise concerns about recurrence due to potential residual disease, prompting oncology teams to enhance surveillance and vigilance for recurrence.

One might also question the relevance of postoperative radiotherapy (PORT): while this technique was not proven effective routinely in the pN2R0 population in the Lung Art trial [95], its use is considered in guidelines, on a case-by-case basis, in the presence of significant risk factors for recurrence [96,97,98]. Indeed, the use of PORT in resected IIIA-N2 remains a very debated topic: although recent series on unselected patients failed to show improvements [99], some wisely selected patients may benefit from it. Some authors suggest the use of risk scores, most of them taking into account the severity of mediastinal involvement [100,101,102]. Thus, HMLN+ patients may be good candidates, as Deng et al. [103] and Guo et al. [104] reported improvements in OS and DFS after adjuvant chemoradation therapy. This point needs more exploration and may be supported by a randomized trial or assessed by a better stratification according to N2 subgroups in retrospective series. To our knowledge, it was not addressed in the Lung Art Trial. Furthermore, current radiation therapy methods (i.e., intensity modulated radiation therapy) may offer better results on disease control and less toxicity, reinforcing their place in adjuvant strategy.

While the relevance of R(un) intersects with the debate on sampling in early stages, its importance in locally advanced stages seems indisputable, a fortiori in the case of HMLN+. The literature provides evidence that the quality of resection and accuracy of lymph node staging are fundamental for the survival of these patients and that a large part of R(un) could be avoided. At a time when conservative approaches, whether in the surgical approach [105] or parenchymal resection [106], are increasingly favored, R(un) reminds us that not all patients are eligible for minimal dissection. The IASLC classification remains underutilized because no therapeutic strategies are attached to it, but this could change in the near future, as it has been advised to integrate it in the upcoming ninth TNM [107].

Ultimately, our findings show that R(un) should be integrated into a broader approach for patient-tailored use of adjuvant therapies, similar to genetic mutation presentation [108], anatomical pathology structure, and TNM staging.

## 5. Conclusions

The uncertain R classification of the IASLC 2005 appears highly relevant, especially in locally advanced stages IIb-IIIA, and helps to discriminate patients with poor prognosis despite being classified as R0 according to previous UICC classifications. Its use in practice remains too limited due to the absence of clear therapeutic strategy changes. Furthermore, its composite nature of R(un) and the rarity of occurrence of some of its criteria prevent the design of prospective trials and series, which is necessary for its legitimacy. The use of this more precise classification would allow for better stratification of recurrence risk and more effective use of adjuvant therapies by thoracic oncology teams.

Finally, we support the idea that Cy+ patients should receive adjuvant chemotherapy, the format of which remains to be determined, while CIS BRM patients could likely benefit from diligent endoscopic surveillance to track local recurrence. Ultimately, HMLN+ patients should be considered at high risk for recurrence, and adjuvant radio-chemotherapy should be discussed.

## Figures and Tables

**Figure 1 cancers-17-01386-f001:**
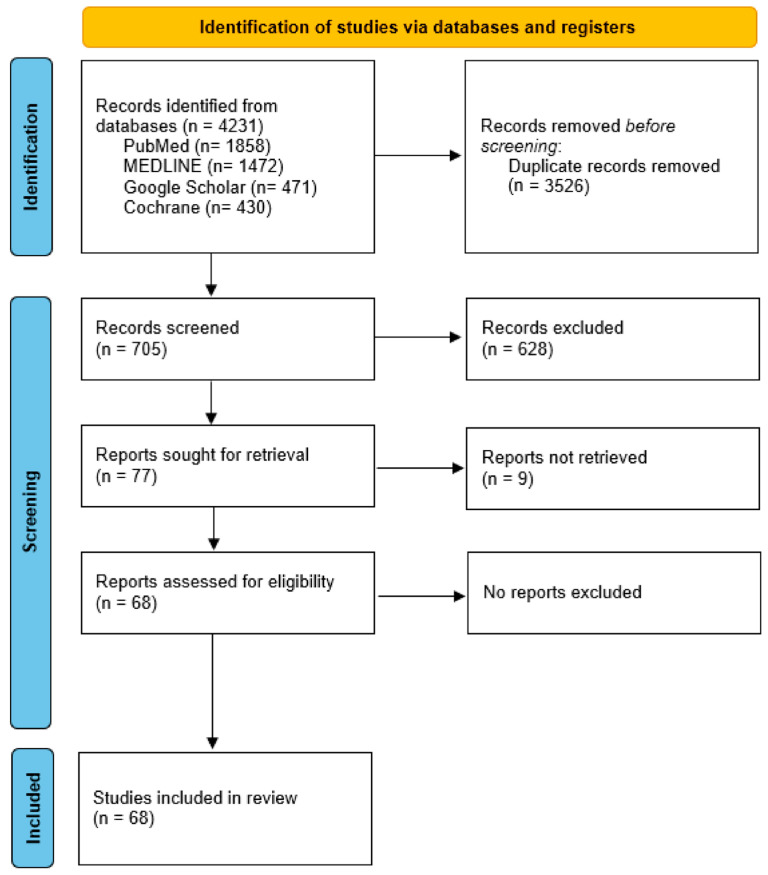
PRISMA flowchart.

**Figure 2 cancers-17-01386-f002:**
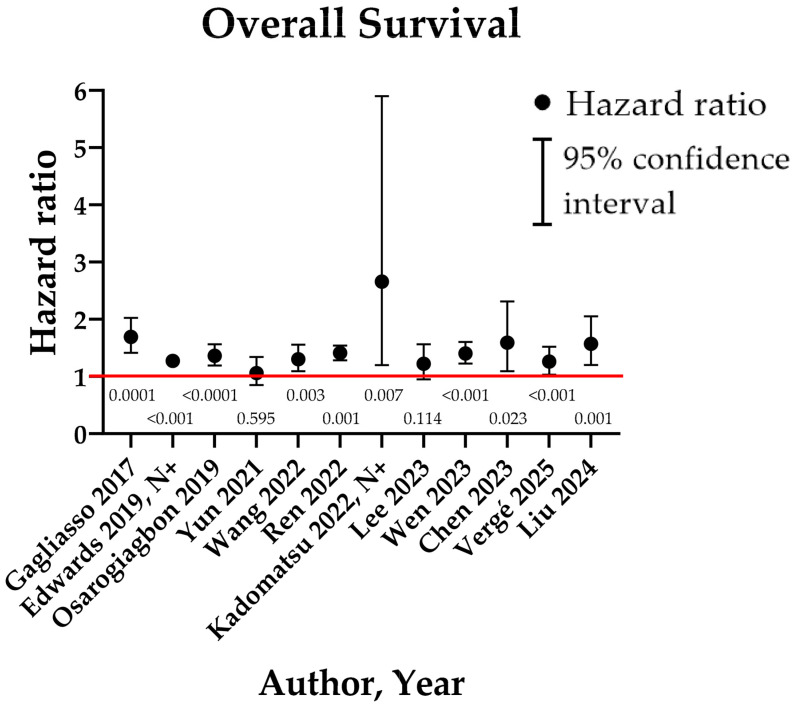
Graphical summary of R(un) group results for overall survival in the literature. Numbers beneath the red line stand for the “*p*” of each study [28,29,30,31,32,33,34,35,36,37,38,39].

**Figure 3 cancers-17-01386-f003:**
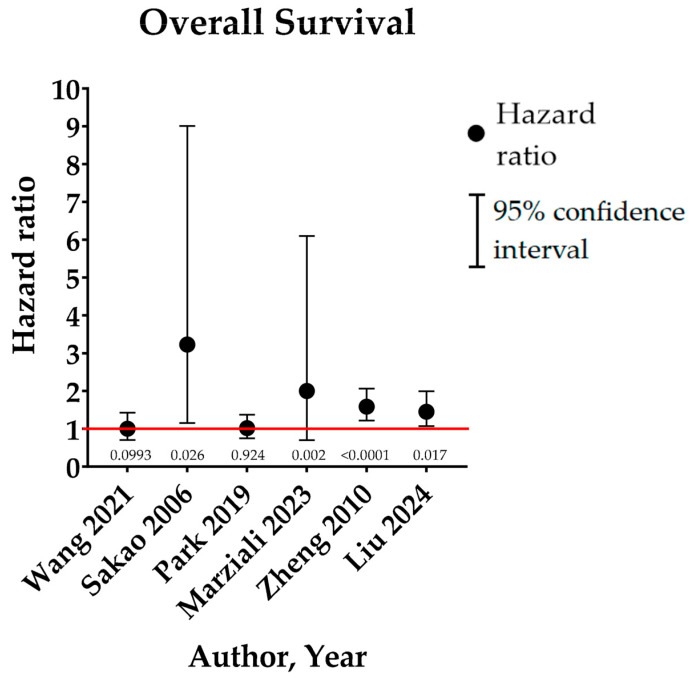
Graphical summary of HMLN+ group results for overall survival in the literature. Numbers beneath the red line stand for the “*p*” of each study. Edwards et al. [29] then utilized the results of a retrospective international cohort, predominantly Japanese and used, notably, for the 8th TNM, for the purpose of validating the IASLC categorization. In this series of 14,712 patients (of whom 14,293 were initially classified as R0, accounting for 97.2% of the total population), 8185 (56%) were reclassified as R(un), of which 8,174 were initially R0. Inadequate lymph node dissection and involvement of the highest lymph node accounted for 95.7% and 3.8% of the reasons for reclassification, respectively. While in the N0 group, the survival of R(un) patients was slightly but significantly reduced (5-year OS: 79% vs. 82%, *p* = 0.04), it appeared significantly impacted in N+ patients (HR 1.27, *p* < 0.001; median OS 50 vs. 70 months; 5-year OS 45% vs. 55%). Finally, there was a small but significant difference within the pT1-2aN0 subgroup (HR 1.22, *p* = 0.0007, 5-year OS 87% versus 83%) that the authors did not explain, but which might correspond to occult N+ involvement and/or skip N2 in a group of patients usually not eligible for adjuvant chemotherapy [41,42,43,44,45,46].

**Table 1 cancers-17-01386-t001:** Published series regarding the IASLC 2005 classification.

			Distribution (%)			Reason for Reclassification (%)				
Author, Year, Country	Patients (n)	Design	R0	R(un)	R1+2	Insufficient LND	HMLN+	CIS BRM	Cy+	HR OS (*p*)
Gagliasso et al. [28], 2017, Italy	pT1-4N0-2M0 (1277)	Retrospective, Single-Institution	1003 (78.5%)	185 (14.5%)	89 (7%)	107 (56.6%)	76 (41%)	5 (2.6%)	NA	1.69 (0.0001)
Edwards et al. [29], 2019, mainly Japan	pT1-4N0-3M0 (14,712)	Retrospective, International Database	6070 (41%)	8185 (56%)	457 (3%)	7824 (95.7%)	312 (3.8%)	11 (0.01%)	34 (0.5%)	N+ 1.27 (< 0.001)
Osarogiagbon et al. [30], 2019, USA	Resected NSCLC (3361)	Retrospective, Population-based	1119 (33%)	2044 (61%)	198 (6%)	2004 (98%)	119 (5.8%)	0 (0%)	3 (0.1%)	1.36 (<0.0001)
Yun et al. [31], 2021, South Korea	pN2 NSLC (1039)	Retrospective, Single-Institution	432 (41.6%)	212 (20.4%)	395 (38%)	21 (10.2%)	185 (89.8%)	6 (from R1)	NA	1.06 (0.595)
Wang et al. [32], 2022, China	pT1-4N0-2M0 (2782)	Retrospective, Single-Institution	1897 (68%)	885 (32%)	0 (0%)	717 (81%)	168 (19%)	0	0	1.302 (0.003)
Ren et al. [33], 2022, China	Resected NSCLC (5293)	Retrospective, Single-Institution	3819 (72.1%)	1371 (25.9%)	103 (1.9%)	929 (67.7%)	511 (37.2%)	2	11	1.41 (0.001)
Kadomatsu et al. [34], 2022, Japan	Resected NSCLC (355)	Retrospective, Single-Institution	197 (55.5%)	158 (44.5%)	0 (0%)	137 (87%)	8 (5%)	3 (2%)	10 (6%)	N+ 2.657 (0.007)
Lee et al. [35], 2023, South Korea	Stage III-N2 (910)	Retrospective, Single-Institution	302 (33.2%)	329 (36.2%)	279 (30.7%)	278 (84.5%)	104 (31.6%)	0	NA	1.22 (0.114)
Wen et al. [36] 2023, China	Resected NSCLC (5200)	Retrospective, Single-Institution	3228 (62%)	1727 (33%)	145 (5%)	1179 (68.3%)	663 (38.3%)	3 (0.2%)	NA	1.4 (<0.001)
Chen et al. [37], 2023, China	Sleeve lobectomy (682)	Retrospective, Single-Institution	489 (71.7%)	110 (16.1%)	83 (12.2%)	28 (25%)	82 (74%)	14 (12%)	0	1.59 (0.023)
Vergé et al. [38], 2025, France	Resected cN0M0 (1108)	Retrospective, Single-Institution	732 (66.1%)	291 (26.2%)	85 (7.7%)	251 (86.3%)	40 (13.7%)	2 (0.6%)	NA	1.26 (<0.001)
Liu et al. [39], 2024, Taiwan	Resected adenocarcinomas (1258)	Retrospective, Single-Institution	429 (33.9%)	829 (65.9)	0 (excluded)	829 (100%)	NA	NA	NA	1.57 (0.001)

**Table 3 cancers-17-01386-t003:** Published series regarding CIS BRM.

Author, Year	Patients (n)	CIS (n)	SCC (n)	CIS % of Population	Local Recurrence (n)	5-Year Survival CIS
Martini et al. [48], 1974	26	2	2	7.6	0	NA
Soorae et al. [49], 1979	434	10	10	2.3	NA	70
Law et al. [50], 1982	1000	9	9	0.9	NA	66.7
Heikkila et al. [51], 1986	1069	5	NA	0.46	NA	NA
Whyte et al. [52], 1988	560	2	2	0.36	NA	NA
Lacasse et al. [53], 1998	399	3	NA	0.8	NA	NA
Ruffini et al. [54], 2004	1090	5	5	0.45	NA	NA
Kawaguchi et al. [47], 2008	4493	9	9	0.2	1	63
Collaud et al. [55], 2009	584	3	3	0.5	2	100
Fernandez et al. [56], 2009	2994	52	45	1.7	NA	NA
Total	15,738	138	114	0.9	/	/

CIS: carcinoma in situ; SCC: squamous cell carcinoma; NA: not available.

## Data Availability

No new data were created or analyzed in this study.

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
