# Peer review of "Uncertain Resection in Lung Cancer: A Comprehensive Review of the International Association for the Study of Lung Cancer Classification"

_cancers, 2025, doi:10.3390/cancers17091386_

Round 1
Reviewer 1 Report
Comments and Suggestions for Authors
The authors analyse the data from various cancer consortium and present the analyzed results and present a schematic evaluation and prospective solutions. However the data is too much heavy in table format and not in figures. In order to present it in journal for wider audience i would suggest the authors present the data in figure format and also keep the language simple for non-specialist audience.
Author Response
Dear reviewer, Thank you for your comments:
the data is too much heavy in table format and not in figures. In order to present it in journal for wider audience i would suggest the authors present the data in figure format and also keep the language simple for non-specialist audience
We added two figures in addition to tables 1 and 2 for more clarity and simplicty. Some points were rewritten, especially objective to be easier to understand.
Reviewer 2 Report
Comments and Suggestions for Authors
- It is good to add numerical results in the abstract. (either in results or conclusion with the abstract)
- The authors should write their aim of the review at the end of the introduction and include a graphical abstract.
- The results are confusing. The authors should add some figures to accomplish the text.
- The main question/problem I have regarding this manuscript is a novelty. The authors summarize some research papers (pages 7-13) to answer what problem.
Author Response
Dear reviewer, Thank you for your comments:
- It is good to add numerical results in the abstract. (either in results or conclusion with the abstract)
- As it is a review and not a meta analysis we have no original numerical to share. Nevertheless we slightly enhanced our abstract.
- The authors should write their aim of the review at the end of the introduction and include a graphical abstract.
- We rewrote our objective more explicitly. Graphical abstract was already providen but was increased.
- The results are confusing. The authors should add some figures to accomplish the text.
- We added two figure in addition of table 1 and 2.
- The main question/problem I have regarding this manuscript is a novelty. The authors summarize some research papers (pages 7-13) to answer what problem.
- Objective was rewritten in both abstract and introduction.
Reviewer 3 Report
Comments and Suggestions for Authors
This review article covers important aspects of future directions and relevance of uncertain resection in lung cancer as a novel strategy to increase patient survival.
This specific strategies compiled in this review article are designed to develop reclassification of lung cancer to help to discriminate patients with poor prognosis and better understand the relevance of the IASLC classification.
The compiled data are supported with informative and important figure and 3 tables. The article concludes with 107 very recent literature references. This consolidated study constitutes crucially important developments, which were never ever reported and in such systematic and specific order and sequences, therefore deserve publication as soon as possible.
The following suggested changes and recommendations should be introduced before the publication of the manuscript:
- Page 5, table 1. Please switch the order of column (Design) with (Patients) column as first. That would enhance the value of the article by validating the number of patients and explore additional citation sequences to benefit the authors.
- Page 15, line 522-3. I propose that the text should be in “bold” because if this unique importance highlighting eligibility/non-eligibility for minimal dissection.
- Page 15, Line 527. Conclusions. This section should be expanded. In its present format, the authors do not fully describe the desired/anticipated prospects of several adjuvant therapies approved by FDA. Authors should also include comparative data available in the literature regarding existing clinical studies demonstrating the high prognostic value of personalized and more effective use and applications depending on category of cancer and number of patients.
The manuscript is of good quality, well-written, and meets the standard for articles published in Cancers. I recommend it for publication after the correction of these minor suggested changes.
Author Response
Dear reviewer,
Thank you for your comments:
- Page 5, table 1. Please switch the order of column (Design) with (Patients) column as first. That would enhance the value of the article by validating the number of patients and explore additional citation sequences to benefit the authors.
- Columns switched
- Page 15, line 522-3. I propose that the text should be in “bold” because if this unique importance highlighting eligibility/non-eligibility for minimal dissection.
- Text is now bold
- Page 15, Line 527. Conclusions. This section should be expanded. In its present format, the authors do not fully describe the desired/anticipated prospects of several adjuvant therapies approved by FDA. Authors should also include comparative data available in the literature regarding existing clinical studies demonstrating the high prognostic value of personalized and more effective use and applications depending on category of cancer and number of patients.
- We added a short paragraph about the potential of personalized adjuvant treatments, thank you for this suggestion
Following other reviewers advice, we reformulated few sentences to make objective of the study more understable. We also added 2 figures in addition to tables 1 and 2 to make them more easy to read. Graphical abstract was simplified.
Round 2
Reviewer 2 Report
Comments and Suggestions for Authors
The authors partially improved their manuscript. They have to add numerical results in the abstract part and add a graphical abstract at the end of the introduction.
Author Response
Dear reviewer,
Thank you for your comments. We added the graphical abstrac below the introduction. We also added numerical in the abstract but as we did not perform a meta analysis we do not have much original statistics to share.

Round 3
Reviewer 2 Report
Comments and Suggestions for Authors
The authors provided the requested improvement and now it is acceptable.